# Complex Precipitates of TiN-MC$_x$ in GCr15 Bearing Steel

**Qianren Tian [1], Guocheng Wang [1,2,3,*], Xinghu Yuan [1], Qi Wang [1,2] and Seetharaman Sridhar [3]**

[1] School of Materials and Metallurgy, University of Science and Technology Liaoning, Anshan 114051, China; tianqianren@126.com (Q.T.); 18341263233@163.com (X.Y.); wangqi8822@sina.com (Q.W.)

[2] Key Laboratory of Chemical Metallurgy Engineering Liaoning Province, University of Science and Technology Liaoning, Anshan 114051, China

[3] Department of Metallurgical and Materials Engineering, Colorado School of Mines, Golden, CO 80401, USA; sseetharaman@mines.edu

\* Correspondence: wang_guocheng@163.com

**Abstract:** Nitride and carbide are the second phases which play an important role in the performance of bearing steel, and their precipitation behavior is complicated. In this study, TiN-MC$_x$ precipitations in GCr15 bearing steels were obtained by non-aqueous electrolysis, and their precipitation mechanisms were studied. TiN is the effective heterogeneous nucleation site for Fe$_7$C$_3$ and Fe$_3$C; therefore, MC$_x$ can precipitate on the surface of TiN easily. The chemistry component of MC$_x$ consists of M$_3$C and M$_7$C$_3$ (M = Fe, Cr, Mn) and Cr$_3$C$_2$. TiN-MC$_x$ with high TiN volume fraction, TiN forms in early stage of solidification, and MC$_x$ precipitates on TiN surface after TiN engulfed by the solidification advancing front. TiN-MC$_x$ with low TiN volume fraction, TiN and MC$_x$ form in late stage of solidification, TiN can not grow sufficiently and is covered by numerous precipitated MC$_x$ particles.

**Keywords:** non-aqueous electrolysis; TiN-MC$_x$; precipitation; bearings; high carbon chromium bearing steel

## 1. Introduction

Controlling microstructures and second phase in steel plays a vital role in the quality of steel. Carbide (M$_3$C, M$_3$C$_2$, M$_7$C$_3$, M = Fe, Cr, Mn) [1,2] and TiN inclusion [3,4] are common second phases in high carbon chromium steel. As a result of good wear resistance and solid solubility with alloy elements (Cr, Mn) [5,6], carbide can retain good mechanical properties of bearing steel during heat treatment [7,8]. Recently, utilization of inclusions has become attractive to improve steel performance. TiN is more harmful to bearing steel than Al$_2$O$_3$ in the same size [9]. Many studies have investigated TiN and Al$_2$O$_3$, MgAl$_2$O$_4$ and MnS, and NbC complex precipitation with inclusion [10–13]. Our previous study [14] found that TiN inclusion was covered by carbide in the etched GCr15 bearing steel metallographic specimens. It is necessary to observe their three-dimensional (3-D) morphologies in steel because the two-dimensional (2D) nature of the particles cannot reflect their real morphologies.

The non-aqueous electrolysis extraction of second phase from steel is an effective method to study its 3D morphologies and composite interfaces. Fang and Ni [15] studied the behaviors of rare earth dissolved in $\alpha$-Fe, Fe-Ce intermetallic compounds and rare earth inclusions via non-aqueous electrolysis. Bi et al. [16] analyzed 3D morphology, quantity, and chemistry of inclusion in ferroalloys by the electrolysis method. Wang et al. [17] observed Al$_2$O$_3$-MnO-SiO$_2$(-MnS) inclusion in steel by non-aqueous electrolysis. Zhang et al. [18,19] analyzed Ca-Mg spinel in cord steel and MnS in heavy rail steel by the electrolysis. Zhang et al. [20] studied the suitable electrolytic conditions for 16MnCrS5 steel.

In this study, 3D morphologies of the carbide (MC$_x$, M = Fe, Cr, Mn) and TiN-MC$_x$ precipitates extracted from GCr15 bearing steel specimens by the non-aqueous electrolysis were observed by

field emission scanning electron microscope-energy dispersive spectrometer (FESEM-EDS). The $MC_x$ chemistry component was confirmed by X-ray diffraction (XRD) and FactSage$^{TM}$ phase diagram calculation. The precipitation mechanism of TiN-$MC_x$ with different volume fraction in GCr15 bearing steels was elucidated.

## 2. Experiment

### 2.1. Chemical Components Analysis

The chemical compositions of GCr15 bearing steel produced by the basic oxygen furnace (BOF)-landle furnace (LF)-vacuum degas (VD)-continuous casting (CC) process in a foundry were determined by direct-reading spectrometer (Model: ARL-3460 Optical Emission Spectrometer, Thermo Fisher Corporation, Waltham, MA, USA). The total oxygen and total nitrogen contents were analyzed using a nitrogen-oxygen analyzer (Model: TC-600, LECO Corporation, St. Joseph, MI, USA). The chemical compositions of the GCr15 bearing steel are shown in Table 1.

**Table 1.** Chemical Compositions of GCr15 Bearing Steel (in mass percent).

| Composition | C | Si | Mn | P | S | Ti | Cr | V | N | Alt | Ca | O (T) |
|---|---|---|---|---|---|---|---|---|---|---|---|---|
| Concentration | 1.01 | 0.25 | 0.36 | 0.012 | 0.0014 | 0.0078 | 1.46 | 0.0099 | 0.0049 | 0.012 | <0.005 | 0.0009 |

### 2.2. Non-Aqueous Electrolysis and XRD Detection

The non-aqueous electrolysis method was used to extract TiN-$MC_x$ particles from the GCr15 bearing steel. Samples with diameter of 10 mm and height of 100 mm were used as anode and copper as cathode. The electrolyte consists of 1% tetramethylammonium chloride, 5% triethanolamine, 5% glycerol, and 89% anhydrous methanol (in volume percentage). The constant voltage direct current (DC) power supply (model: DH1720A-1) was used to keep the current density between 40–60 mA/cm$^2$. The temperature of the electrolyte was kept at 268–278 K (−5–5 °C). Argon gas was used to stir organic electrolyte. After electrolysis, steel samples were placed in a beaker containing ethanol and vibrated with ultrasonic wave to separate all particles from the samples surface. $MC_x$ and inclusions in ethanol were further separated by the magnetism. The inclusion particles were transferred directly to the double-sided carbon bands attached to the conductive material and then were observed by FESEM-EDS. After magnetic separation, $MC_x$ was analyzed by XRD (Model: X'Pert Powder, Malvern PANalytic Ltd., Malvine, UK, the detection parameters are that Cu K$\alpha\lambda$ = 0.154178 nm, tube current 40 mA and tube voltage 40 kV, scanning scope 30–85 °C, step length 0.013 s, residence time 5 s).

## 3. Result

### 3.1. Observation for Particles

2D morphologies of TiN-$MC_x$ in the metallographic specimens etched by 4% nitric acid alcohol were observed by FESEM-EDS and are shown in Figure 1. The EDS points are the black crosses, and the analysis for elements can both be seen in Figure 1. The dark grey particles are TiN inclusions, and the light grey particles are $MC_x$. Figure 1a shows a long strip and large size TiN with a small amount of $MC_x$ around it. Figure 1b–d show TiN with less pronounced aspect ratios and it is covered by a larger number of $MC_x$, which, in some cases, form a continuous layer rather than discrete particles.

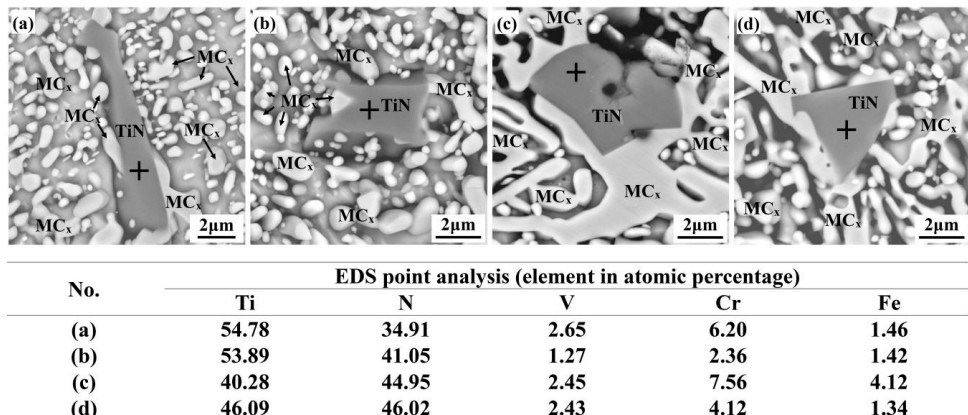

| No. | EDS point analysis (element in atomic percentage) | | | | |
|-----|------|------|------|------|------|
|     | Ti    | N     | V    | Cr   | Fe   |
| (a) | 54.78 | 34.91 | 2.65 | 6.20 | 1.46 |
| (b) | 53.89 | 41.05 | 1.27 | 2.36 | 1.42 |
| (c) | 40.28 | 44.95 | 2.45 | 7.56 | 4.12 |
| (d) | 46.09 | 46.02 | 2.43 | 4.12 | 1.34 |

**Figure 1.** TiN-MC$_x$ particles and energy dispersive spectrometer (EDS) point analysis for TiN part in etched metallographic specimens. (**a**) Long-strip and large size TiN with MC$_x$; (**b**)–(**d**) small size TiN with MC$_x$.

3D morphologies, chemistries of TiN-MC$_x$ and EDS point analysis for TiN part were also observed by FESEM-EDS; the atomic percentage of element can be seen in each element mapping. Figure 2a shows TiN-MC$_x$ with large size TiN inclusion (in comparison to MC$_x$) whose size was approximately 25 μm. This category of TiN-MC$_x$ is denoted as "TiN-MC$_x$ with high TiN volume fraction". The elements mapping shows that Ti and V can form the solid solution; however, the metallic elements in MC$_x$ are Fe and Cr. Mn cannot be detected because of its low content. Figure 2b shows that TiN-MC$_x$ with small size TiN are approximately 5 μm, with its shape more closely resembling a sphere. In comparison with the TiN-MC$_x$ in Figure 2a, TiN in the precipitates is clearly smaller. This category of TiN-MC$_x$ is denoted as "TiN-MC$_x$ with low TiN volume fraction". The TiN-MC$_x$ with low TiN volume fraction is almost completely covered by MC$_x$.

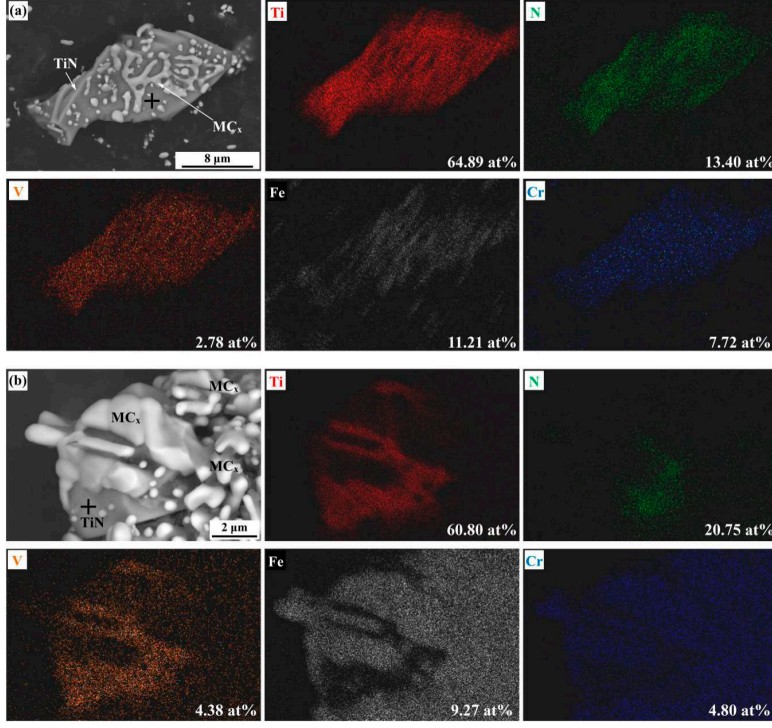

**Figure 2.** Morphologies and elements mapping of TiN-MC$_x$ and EDS point analysis for TiN part. (**a**) High TiN volume fraction; (**b**) low TiN volume fraction.

Figure 3 shows the 3D morphologies of $MC_x$ and elements mapping of Fe, Cr, and Mn. The morphologies of single $MC_x$ particles are not significantly different from that of $MC_x$ on the surface of TiN inclusion. Figure 3a shows a spherical $MC_x$ with a smaller size, less 1 μm. Figure 3b shows a flat $MC_x$ with approximately 1 μm, whereas a $MC_x$ with a shape of a long strip of length 6 μm can be seen in Figure 3c, which is rarely found in steel. Figure 3d shows $MC_x$ transferred on the conductive carrier, and the $MC_x$ are predominantly spherical and flat. Figure 3e,f are cluster-like $MC_x$, with Fe, Cr and Mn elements mapping results. The size of cluster-like $MC_x$ are approximately 15 μm. However, the cluster-like $MC_x$ exhibits the morphology of banded or reticulated $MC_x$ in metallographic samples. In Figure 3e,f, C is not shown because $MC_x$ and inclusions were transferred on the carbon bands.

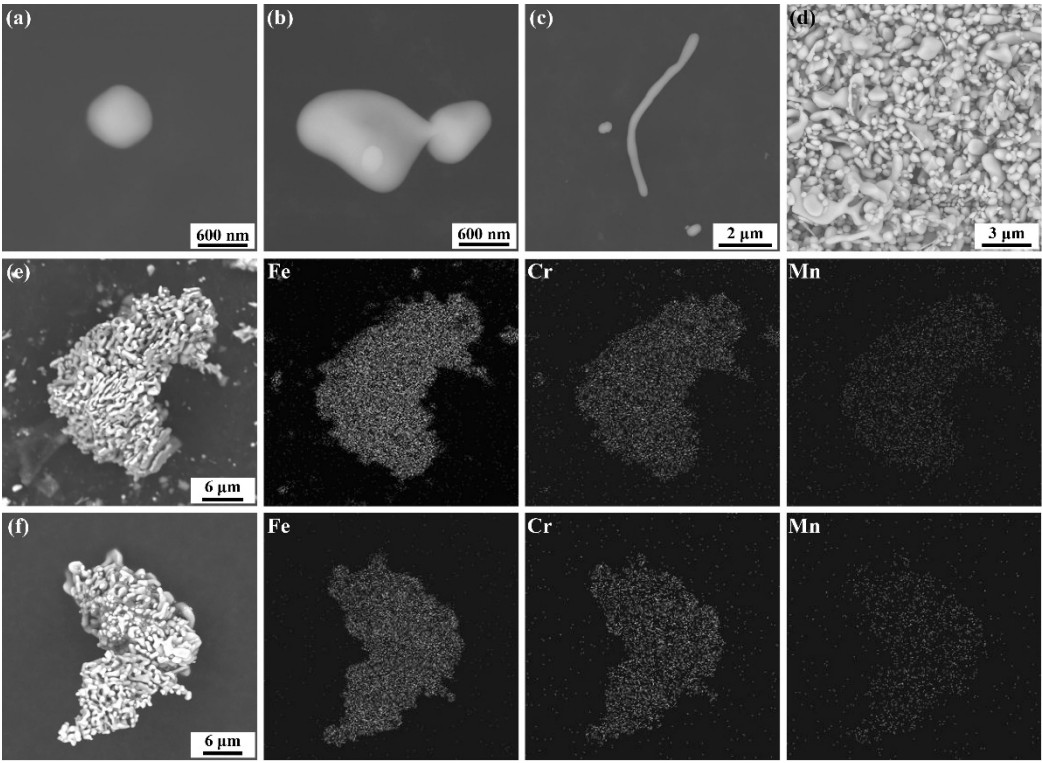

**Figure 3.** Morphologies and elements mapping of $MC_x$. (**a**) Ball-like $MC_x$; (**b**) flat-like $MC_x$; (**c**) long stripe-like; (**d**) $MC_x$ transferred on the conductive carrier; (**e**) and (**f**) cluster-like $MC_x$ with elements mapping of Fe, Cr, and Mn.

*3.2. XRD Result*

Figure 4 shows the result of the $MC_x$ XRD experiment; the structures of the $MC_x$ are predominantly $M_3C$ and $M_7C_3$, with $M_3C$ as the dominant carbide. At $2\theta = 48.6°$, several $Cr_3C_2$ remained. The results are similar to carbides in GCr15 bearing steel after electroslag remelting-continuous casting (ESR-CC) process, as demonstrated by Du et al. [2] The main $MC_x$ in that study were $M_3C$, $M_3C_2$, and $M_7C_3$, and the content of Cr in their sample was 1.47% and 0.31% [2], respectively, which is similar to that seen in our steel.

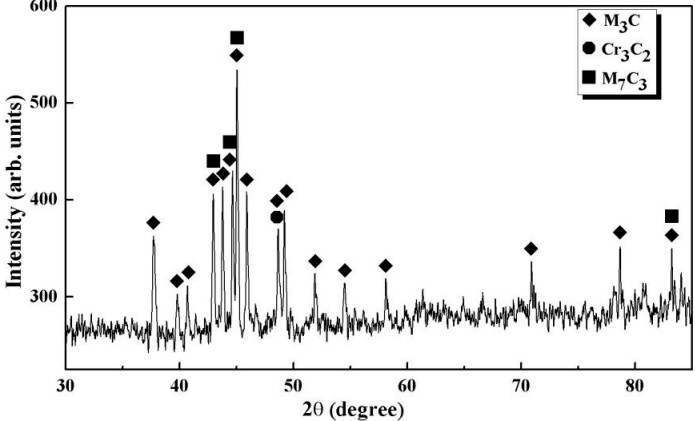

**Figure 4.** XRD analysis result of the extracted $MC_x$ in the GCr15 bearing steel.

## 4. Discussion

### 4.1. Thermodynamic Analysis

The phase diagram of Fe-1.5%Cr-C system was calculated via FactSage$^{TM}$ 7.2 thermodynamic software and steel database [21]. The calculated phase diagram for the conditions, [pct Cr] = 1.5, [pct C] = 0.5–1.5 ([pct element] is mass percent of the element in steel), and the temperature range from 298 K (25 °C) to 1873 K (1600 °C), is shown in Figure 5 and the gray part presents the mushy zone of steel. When [pct C] = 1, equilibrium transformation of steel is that liquid → FCC + liquid → FCC → $M_3C$ + FCC → $M_3C$ + FCC + BCC → $M_3C$ + BCC → C + $M_3C$ + BCC → C + BCC + $M_7C_3$ → C + BCC + $Cr_3C_2$. The liquidus temperature and solidus temperature are close to the calculated values in our previous paper [liquidus and solidus temperature are 1723 K (1450 °C) and 1601 K (1328 °C), respectively] [14]. When the temperature is slightly lower than 1173 K (900 °C), $M_3C$ gradually precipitates from FCC phase; when the temperature is approximately 913 K (640 °C), $M_3C$ gradually transforms to $M_7C_3$. When the temperature is slightly higher than 773 K (500 °C), the carbide gradually transforms into $Cr_3C_2$. During this process, phase transformation will be difficult to be completed to the phase fraction, dictated by the equilibrium phase diagram which leads to the transition layers. The main phases formed during temperature gradual decreasing are $M_3C$, $M_7C_3$, $Cr_3C_2$, and their content decreases in turn. FactSage$^{TM}$ calculation results are consistent with XRD results, in which $MC_x$ were found to be $M_3C$, $M_7C_3$, and $Cr_3C_2$.

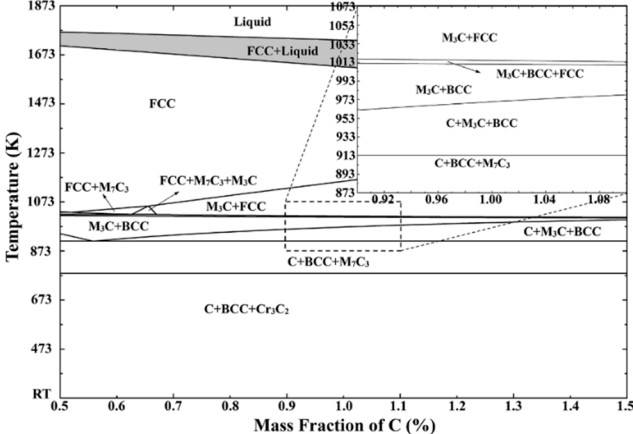

**Figure 5.** Phase diagram for Fe-1.5%Cr-C system (the shadow part is the mushy zone of steel, C presents the pure substance C(s); $M_3C$ (Cementite) presents $Fe_3C$ with dissolved Cr, Mn; $M_7C_3$ presents carbide phase found in Cr, Mn-containing steels; FCC and BCC present the face-centered cubic iron (γ-Fe) and body-centered cubic iron (α-Fe), respectively).

As demonstrated in previous work [14,22], TiN precipitates in the mushy zone of GCr15 bearing steel, and their size is affected by the concentration of Ti and N around TiN crystal nucleus. Ti and N both are positive segregation elements ($k > 0$), their concentrations and, consequently, the supersaturation increases with an increase in solid fraction, and TiN precipitation become easier during solidification process. Recently, Li et al. [23] studied the precipitation of TiN inclusions in GCr15 bearing steel during solidification by calculating the thermodynamics and growth kinetics in consideration of solidification segregation of the solute elements. They believe that the effect of Ti content on the size of TiN inclusions is greater than that of N content.

Fe, Cr, Mn, and C can precipitate on TiN, which is the heterogeneous nucleation site for $MC_x$. It can be seen in Figure 2a that the size of TiN is large and, consequently, TiN would be formed at the early stage of solidification and has enough time to grow. TiN in Figure 2b precipitates at the late stage of solidification. The diffusion coefficients $D_i$, $i$ = Cr, Ti, Mn, C, N in γ-phase were calculated according to the phase diagrams, when [pct C] = 1 and FCC precipitates at 1673 K (1400 °C). The relationship between diffusion coefficients and temperature from 1673 to 1173 K (1400 to 900 °C) is shown in Figure 6a, and the parameters are shown in Table 2. From Figure 6a, in 1673–1642 K (1400–1369 °C), $D_N^γ > D_C^γ > D_{Ti}^γ > D_{Mn}^γ > D_{Cr}^γ$; in 1642–1173 K (1369–900 °C), $D_C^γ > D_N^γ > D_{Ti}^γ > D_{Mn}^γ > D_{Cr}^γ$. The results indicate that the diffusion of C and N plays a dominant role. The diffusion of C is more efficient than N after the temperature has decreased under 1642 K. In contrast, the diffusion of Cr, Ti, and Mn are extremely small in γ-phase; the content of Cr is much larger than that of Mn and Ti, which would not affect the precipitation of $MC_x$. That indicates that TiN may precipitate more easily in the range of temperature of 1673 K–1642 K in comparison with $MC_x$, which precipitates easily during the following decreasing temperature process.

**Table 2.** Diffusion coefficient and equilibrium partition coefficient of C, Cr, Mn, Ti, and N in γ-phase. Data from [3,24–26].

| Element | Equilibrium Partition Coefficient, $k$ | Diffusion Coefficient in γ-phase (cm²/s) |
|---|---|---|
| C | 0.34 | 0.0761·EXP (−134600/RT) |
| Cr | 0.85 | 0.0012·EXP (−219000/RT) |
| Mn | 0.78 | 0.486·EXP (−276100/RT) |
| Ti | 0.33 | 0.15·EXP (−251000/RT) |
| N | 0.48 | 0.91·EXP (−168500/RT) |

The segregation degree of C, Cr, Ti, Mn, and N during solidification is calculated according to Equations (1) to (5) [27] in Figure 6b, at cooling rate of 0.5 K/s, temperature of 1723 K to 1601 K (1450 °C to 1328 °C) and corresponding solid fraction is 0–1. The order of segregation from high to low is Ti, C, N, Mn, and Cr at the same solid fraction. The segregation degree of C at the late stage of solidification reaches tens of times the initial content. Therefore, $MC_x$ precipitation on the TiN surface becomes easier at the late stages of solidification.

$$\frac{[\text{pct X}]_t}{[\text{pct X}]_0} = \left[1 - \left(1 - \frac{\beta k_i}{1+\beta}\right)\cdot f_s\right]^{\frac{k_i-1}{1-\frac{\beta k_i}{1+\beta}}} \tag{1}$$

$$\beta = \frac{4D_i^γ}{L^2} \tag{2}$$

$$\tau = \frac{T_l - T_s}{R_c} \tag{3}$$

$$L = 143.9 \times R_c^{-0.386}, \left([\text{pctC}] = 1\right) \tag{4}$$

$$T = T_{Fe} - \frac{T_{Fe} - T_l}{1 - f_s\frac{T_l-T_s}{T_{Fe}-T_s}} \tag{5}$$

Here, [pct X]$_t$is the C concentration at solidification front, [pct X]$_0$ is the initial C concentration; $f_s$ is solid fraction; $k_i$ is equilibrium distribution coefficient of C, Cr, Mn, Ti, and N in γ-phase; $D_i^\gamma$ is diffusion coefficient of C, Cr, Mn, Ti, and N in γ-phase, cm$^2$/s; $\tau$ is the local cooling time, s; R$_c$ is the local cooling rate, K/s; *L* is secondary arm space, μm; T$_{Fe}$, T$_l$, and T$_s$ are the melting point of pure iron [1809 K (1536 °C)], the liquidus temperature [1723 K (1450 °C)], and the solidus temperature [1601 K (1328 °C)] of GCr15 steel [14], respectively.

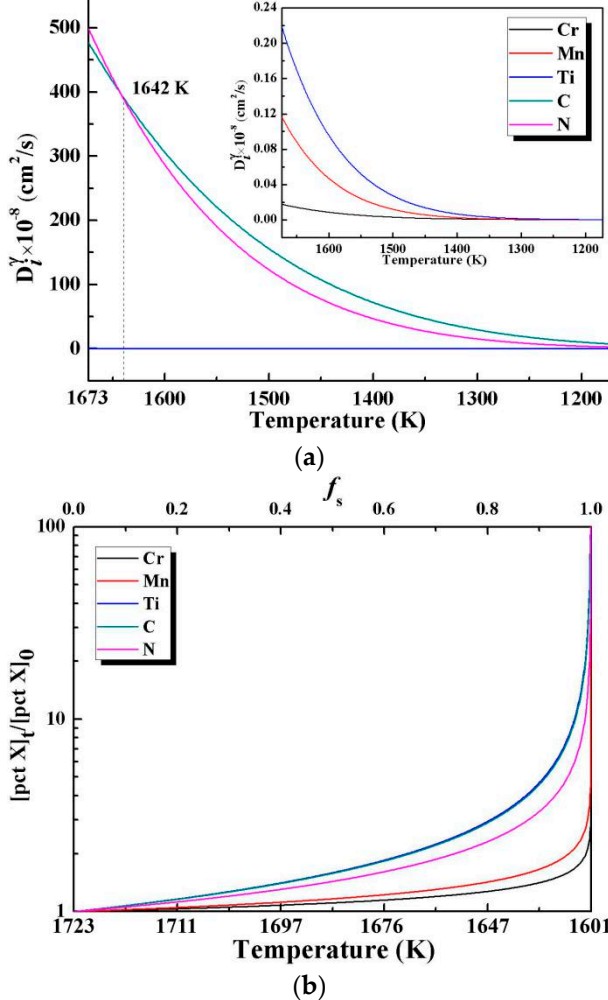

**Figure 6.** Diffusion coefficient change in γ-phase and segregation degree of C, Mn, Ti, C, N during solidification process (**a**) diffusion coefficient change; (**b**) segregation degree.

*4.2. Crystallographic Analysis*

Using the disregistry theory [28,29], the inconsistency of lattice parameters between matrix and nucleating phase can indicate the effectiveness of nucleating catalysts. Choosing three crystal planes and three crystal orientations of the matrix and new phase crystal, the corresponding crystal parameters can calculate the disregistries between two phases by Equation (6). Because M$_3$C and M$_7$C$_3$ are substitutional solid solutions (Cr and Mn take the position of Fe in carbides) [30], the minimum disregistries of TiN-M$_7$C$_3$, TiN-M$_3$C, and Fe$_3$C-Fe$_7$C$_3$ were verified by the parameters of TiN [14], Fe$_7$C$_3$ [31] and Fe$_3$C [32]. The parameters and calculated results are shown in Table 3, and the disregistries diagram is shown in Figure 7. The disregistry between [$\bar{1}$11] (110)TiN‖ [$\bar{1}$11] (110)Fe$_7$C$_3$, [$\bar{1}$11] (110)TiN‖ [$\bar{1}$11] (110)Fe$_3$C and [011](100)Fe$_3$C‖[011](100)Fe$_7$C$_3$ is 1.49%, 5.16% and 7.40%, respectively. The results

show that the disregistries between TiN and $Fe_7C_3$ and $Fe_3C$ are both small; consequently, TiN would provide suitable heterogeneous nucleation sites for $Fe_7C_3$ and $Fe_3C$.

$$\delta^{(hkl)_s}_{(hkl)_n} = \frac{1}{3}\sum_{i=1}^{3} \frac{\left| \left( d_{[uvw]^i_s} \cdot \cos\theta \right) - d_{[uvw]^i_n} \right|}{d_{[uvw]^i_n}} \times 100\% \tag{6}$$

where $\delta^{(hkl)_s}_{(hkl)_n}$ is disregistry between a solid plane $(hkl)_s$ and a nucleate plane $(hkl)_n$; $d_{[uvw]^i_s}$ and $d_{[uvw]^i_n}$ are the interatomic spacing along a low-index direction $[uvw]_s$ and the interatomic spacing along a low-index direction $[uvw]_n$; $\theta$ is the angle between $[uvw]_s$ and $[uvw]_n$.

**Table 3.** Parameters and lattice disregistry between tin and nucleation phase.

| Substance (Space Group) | Lattice Parameters (Length Unit: Å) | | | | |
|---|---|---|---|---|---|
| | **a** | **b** | **C** | $\alpha = \beta = \gamma$ (°) | |
| $Fe_7C_3$ (Pnma) [31] | 4.537 | 6.892 | 11.913 | 90 | |
| **TiN-$Fe_7C_3$** | $[hkl]_s$ | $[hkl]_n$ | $d_{[hkl]_s}$ | $d_{[hkl]_n}$ | $\theta$ (°) | Disregistry |
| (100)TiN‖(100)$Fe_7C_3$ | [001] | [001] | 2.118 | 11.913 | 0(-) | |
| | [011] | [011] | 2.995 | 13.763 | 14.949 | 6.52% |
| | [010] | [010] | 2.118 | 6.892 | - | |
| (110)TiN‖(110)$Fe_7C_3$ | [$\bar{1}$10] | [001] | 2.995 | 11.913 | - | |
| | [$\bar{1}$11] | [$\bar{1}$11] | 3.668 | 14.492 | 0.556 | 1.49% |
| | [001] | [$\bar{1}$10] | 2.118 | 8.251 | - | |
| (111)TiN‖(111)$Fe_7C_3$ | [0$\bar{1}$1] | [0$\bar{1}$1] | 2.995 | 13.763 | 5.275 | |
| | [$\bar{1}$01] | [$\bar{1}$01] | 2.995 | 12.748 | 18.715 | 8.93% |
| | [$\bar{1}$10] | [$\bar{1}$10] | 2.995 | 8.251 | - | |
| **Substance (Space Group)** | **Lattice Parameters (Length Unit: Å)** | | | | |
| | **a** | **b** | **C** | $\alpha = \beta = \gamma$ (°) | |
| $Fe_3C$ (Pnma) [32] | 5.092 | 6.741 | 4.527 | 90 | |
| **TiN-$Fe_3C$** | $[hkl]_s$ | $[hkl]_n$ | $d_{[hkl]_s}$ | $d_{[hkl]_n}$ | $\theta$ (°) | Disregistry |
| (100)TiN‖(100)$Fe_3C$ | [001] | [001] | 2.118 | 4.527 | - | |
| | [011] | [011] | 2.995 | 8.120 | 11.116 | 6.92% |
| | [010] | [010] | 2.118 | 6.741 | - | |
| (110)TiN‖(110)$Fe_3C$ | [$\bar{1}$10] | [$\bar{1}$10] | 2.995 | 8.448 | - | |
| | [$\bar{1}$11] | [$\bar{1}$11] | 3.668 | 9.585 | 26.551 | 5.16% |
| | [001] | [001] | 2.118 | 4.527 | - | |
| (111)TiN‖(111)$Fe_3C$ | [0$\bar{1}$1] | [0$\bar{1}$1] | 2.995 | 8.448 | 3.227 | |
| | [$\bar{1}$01] | [$\bar{1}$01] | 2.995 | 8.120 | 8.258 | 9.25% |
| | [$\bar{1}$10] | [$\bar{1}$10] | 2.955 | 6.813 | - | |
| **$Fe_3C$-$Fe_7C_3$** | $[hkl]_s$ | $[hkl]_n$ | $d_{[hkl]_s}$ | $d_{[hkl]_n}$ | $\theta$ (°) | Disregistry |
| (110)$Fe_3C$‖(110)$Fe_7C_3$ | [001] | [001] | 4.527 | 11.913 | - | |
| | [$\bar{1}$11] | [$\bar{1}$11] | 9.585 | 14.492 | 27.107 | 11.38% |
| | [$\bar{1}$10] | [$\bar{1}$10] | 8.448 | 8.251 | - | |
| (100)$Fe_3C$‖(100)$Fe_7C_3$ | [001] | [001] | 4.527 | 11.913 | - | - |
| | [011] | [011] | 8.120 | 13.763 | 26.068 | 7.40% |
| | [010] | [010] | 6.741 | 6.892 | - | - |
| (111)$FeC_3$‖(111)$Fe_7C_3$ | [$\bar{1}$01] | [0$\bar{1}$1] | 8.120 | 13.763 | 2.983 | |
| | [0$\bar{1}$1] | [$\bar{1}$01] | 8.448 | 12.748 | 15.488 | 21.0% |
| | [$\bar{1}$10] | [$\bar{1}$10] | 6.813 | 8.251 | - | |

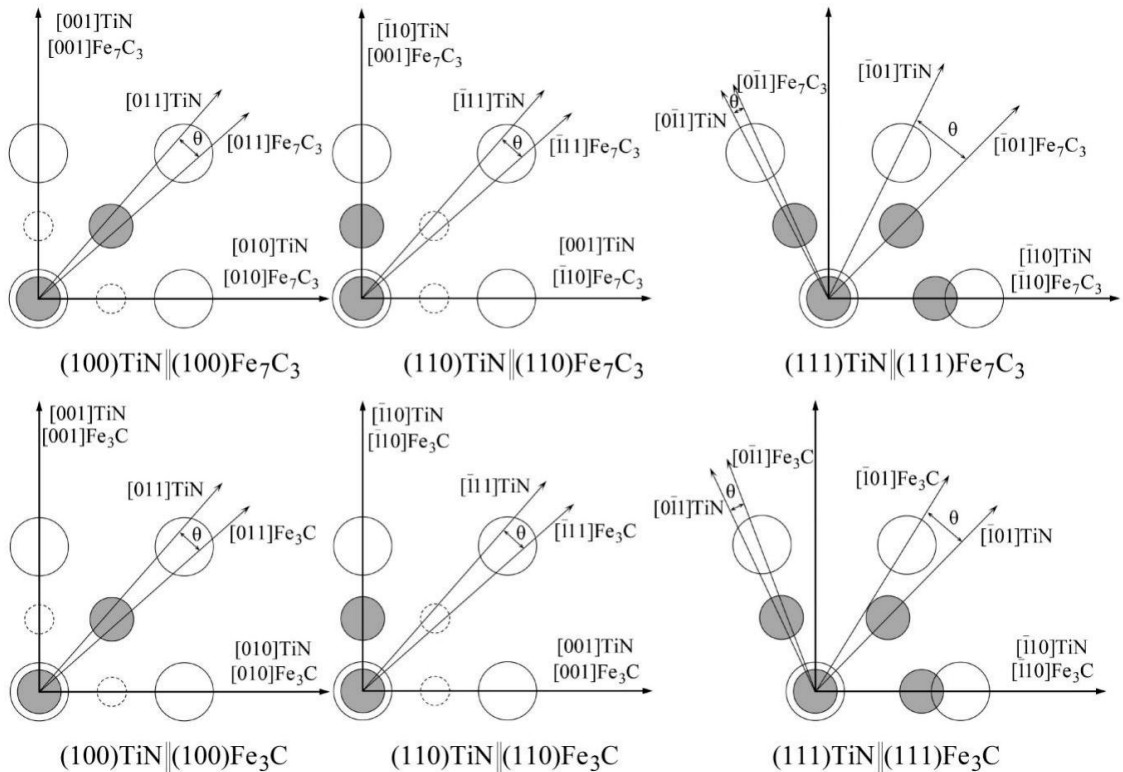

**Figure 7.** Schematic diagram of disregistry between TiN and FeC$_x$ ($x$ = 3/7, 1/3).

### 4.3. Pushing and Engulfment Behavior of Particles

Descotes et al. [33] found that TiN particles could be engulfed by the solid side at the solid-liquid interface in the solidification process. However, Pervushuin et al. [34] reported that TiN was pushed into the liquid side in molten steel during solidification. In our previous study [22], the local cooling rate and movement velocity of solidification front were confirmed as 0.7 K/s and 3 μm/s by the observation of confocal laser scanning microscope (CLSM), respectively. The changes of temperature, time, and distance are approximately 7 K, 10.6 s, and 32 μm, as shown in Figure 8. The critical velocity of pushing or engulfment $V_{cr}$ = 23/$R$ ($R$ is the radius for globular particles) [35], when $R$ is 12.5 μm and 2.5 μm (the particles in Figure 2), $V_{cr}$ are equal to 1.84 μm/s and 9.2 μm/s, respectively. This indicates that the large size TiN inclusion is easier to be engulfed than small size TiN. In the actual process, the local cooling rate is 0.5 to 10 K/s. The higher the local cooling rate is, the faster the solidification front moves, and the more easily the inclusions are engulfed.

The size of TiN-MC$_x$ is larger than TiN, TiN-MC$_x$ moves more slowly than TiN, and is easier to be swallowed by the solidification front. After the engulfment, particles will continue to grow through solid state diffusion, the rate of which will decrease with decreasing temperature. For TiN-MC$_x$ pushed to the liquid phase, elements segregation provides the possibility for the growth of MC$_x$ on TiN, TiN-MC$_x$ growing until its size is large enough to be engulfed by the solid phase. The precipitation mechanism of TiN-MC$_x$ in different solidification periods can be confirmed.

- TiN-MC$_x$ with high TiN volume fraction precipitates at the early stage of solidification and has better growth kinetics in the melt. After being engulfed by the solidification front, MC$_x$ grows at a lower rate on the surface of TiN.
- TiN-MC$_x$ with low TiN volume fraction precipitates in the late stage of solidification and does not have enough time to grow to large size. Due to high C concentration and segregation, a large amount of MC$_x$ precipitates on TiN surface. When TiN-MC$_x$ is large enough and engulfed by the solidification front, the volume fraction of MC$_x$ is large enough to cover the TiN particle.

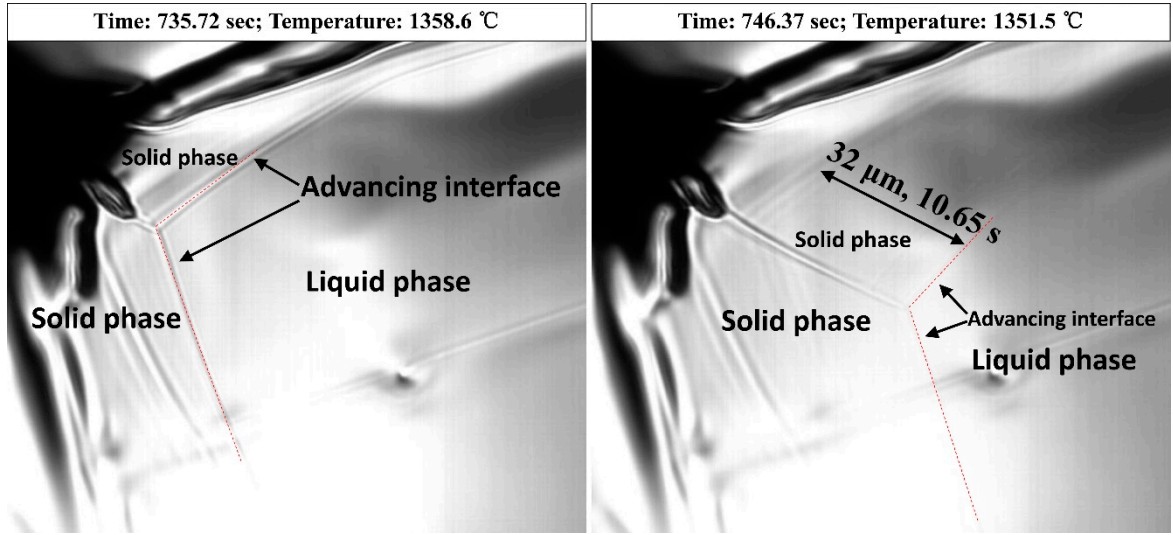

**Figure 8.** Parameters and distance changes of solidification front in GCr15 bearing steel.

## 5. Conclusions

In this study, TiN-MC$_x$ precipitation mechanism in GCr15 bearing steels were analyzed by combining the experiments of non-aqueous electrolysis, FESEM observation with EDS and XRD analysis as well as with the theoretical analysis of thermodynamic and crystallographic analyses in addition to CLSM observation for pushing and engulfment behavior of particles. The main conclusions can be drawn:

(1)   TiN-MC$_x$composed of TiN and MC$_x$, TiN is the effective heterogeneous nucleation site for Fe$_7$C3 and Fe$_3$C, in which the MC$_x$ precipitates on the surface of TiN was observed in GCr15 bearing steel.

(2)   MC$_x$(M = Fe, Cr, Mn) in GCr15 bearing steel smelted by converter is mainly composed of M$_3$C, M$_7$C$_3$, and Cr$_3$C$_2$.

(3)   TiN-MC$_x$ with high TiN volume fraction precipitates at the early solidification stage. After being engulfed by the solidification front, MC$_x$ grows at a lower rate on the surface of TiN.

(4)   TiN-MC$_x$with low TiN volume fraction precipitates in the late solidification stage and does not have enough time to grow to large size. When the size of TiN-MC$_x$ is large enough and is engulfed by the solidification front, the volume fraction of MC$_x$ is large enough to cover TiN particle because of high C concentration and segregation.

**Author Contributions:** Methodology, G.W. and X.Y.; Project administration, Q.W.; Supervision, G.W. and S.S.; Writing—original draft preparation, Q.T.; Writing—review and editing, Q.T.; Supervision, G.W.

**Funding:** This research was funded by National Natural Science Foundation of China (Grant Nos. 51874170 and 51634004).

**Conflicts of Interest:** The authors declare no conflict of interest.

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
