# Peer review of "Complex Precipitates of TiN-MCx in GCr15 Bearing Steel"

_metals, doi:10.3390/met9060641_

Reviewer 1 Report

The combination of the experiments of non-aqueous electrolysis, FESEM observation with EDS, XRD analysis, theoretical thermodynamic and crystallographic analyses, and CLSM observation for pushing and engulfment behavior of particles give reliable enaugh arguments to support the conclussions drawn.

The manuscript is well organized and provides novelties in the field of kinetics and thermodynamics of precipitates.

Author Response

C.1 The combination of the experiments of non-aqueous electrolysis, FESEM observation with EDS, XRD analysis, theoretical thermodynamic and crystallographic analyses, and CLSM observation for pushing and engulfment behavior of particles give reliable enough arguments to support the conclusions drawn.

A.1: Thanks for your comments.

C.2 The manuscript is well organized and provides novelties in the field of kinetics and thermodynamics of precipitates.

A.2:Thanks for your points in favor.

Reviewer 2 Report

The manuscript could be accepted after revision.

Some required information are missed:

1)     XRD spectra are displayed without specifying wavelength used in the experiments, database and file for specific compounds identification. These information are needed.

2)     Observation of particles has been performed through SEM. Images and chemical maps are reported. However the maps are not clear. Where it is visible that: “Ti and V can form the solid solution”? (line 85). To better understand the elements of each particles you should perform EDS point analysis inside the particles.

3)     You can not report a XRD spectrum without the identified phase. Which kind of carbides are M3C and M7C3? Without compounds identification the spectrum in fig 4 has not sense.

Author Response

Q.1 XRD spectra are displayed without specifying wavelength used in the experiments, database and file for specific compounds identification. These information are needed.

A.1: Yes, it is a necessary part to show the detail parameters of XRD detection. The parameters have been added in line 72 and 73 of the revised manuscript.

Q.2 Observation of particles has been performed through SEM. Images and chemical maps are reported. However the maps are not clear. Where it is visible that: “Ti and V can form the solid solution”? (line 85). To better understand the elements of each particles you should perform EDS point analysis inside the particles.

A.2: Yes, the EDS point analysis is shown in Figs 1 and 2, the black cross is the position of EDS point, and the atomic percentage of elements can be seen in the revised Figs 1 and 2, the corresponding description is added in line 77 and 78 of the revised manuscript.

Q.3 You can not report a XRD spectrum without the identified phase. Which kind of carbides are M3C and M7C3? Without compounds identification the spectrum in fig 4 has not sense.

A.3: M3C and M7C3 are the solid solution carbides of the Fe3C, Fe7C3 and the elements Cr and Mn, they can be detected as Fe3C, Fe7C3, Fe2.7Mn0.3C, Fe1.8Mn1.2C, Cr7C3, and Cr3C2, the search and match parameters can be seen as pictures below. Although, the total number of patterns is 314,484, there might be some compounds ignored due to the limited database, and there is no identified phase about the solid solution carbides in the database. Therefore, based on the SEM-EDS results and the others’ studies we considered the M3C and M7C3 are (Fe, Cr, Mn)3C, and (Fe, Cr, Mn)7C3, respectively, in converter smelt process. So, the value of the intensity is added in the revised Figure 4 to character the corresponding compounds of the carbides in GCr15 bearing steel.

Reviewer 3 Report

Review for metals-511278

Complex Precipitates of TiN-MCx in GCr15 Bearing Steel

The authors address an interesting research topic for the journal Metals. It is a rigorous and well-organized paper. Anyway, some recommendations should be considered:

Comments and Suggestions for Authors

[1]        Please, include “bearings” in the following sentence: Line 27 “high carbon chromium bearing steel”

[2]        Please, improve the visibility of some figures (e.g. Figs. 5, 6 and 7 are not clear).

[3]        Try aligning the rows in Table 3 correctly.

[4]        Please, adjust the "Disregistry " column to include such a title in a single line.

[5]        Please, revise the format of the references 34 y 35.

[6]   Although the number and the selection of references is adequate, it would be advisable to include some papers from the journals of MDPI editorial (Materials, Metals, Applied Sciences, etc.) related to the topic of the manuscript.

Author Response

Q.1 Please, include “bearings” in the following sentence: Line 27 “high carbon chromium bearing steel”

A.1: Yes, it has been added in line 22 of the revised manuscript.

Q.2 Please, improve the visibility of some figures (e.g. Figs. 5, 6 and 7 are not clear).

A.2: Yes, the visibility of figures have been improved, Figs. 5, 6, 7 can be seen below:

Q.3 Try aligning the rows in Table 3 correctly.

A.3: Row spacing of Table 3 has been changed in the revised manuscript. 

Q.4 Please, adjust the "Disregistry " column to include such a title in a single line.

A.4: Yes, it has been revised in Table 3.

Q.5 Please, revise the format of the references 34 y 35.

A.5: Yes, the format of the references 34 and 35 has been modified, the revised part can be seen in line 335 and 337 of the revised manuscript.

Q.6 Although the number and the selection of references is adequate, it would be advisable to include some papers from the journals of MDPI editorial (Materials, Metals, Applied Sciences, etc.) related to the topic of the manuscript.

A.6: Yes, a necessary paper published in Materials has been added in the revised manuscript as the cited references 23, which is in line 149-152 of the revised manuscript.

[23] Li, B.; Shi, X.; 1,2, Guo H. J.; Guo, J.; Study on Precipitation and Growth of TiN in GCr15 Bearing Steel during Solidification. Materials 2019, 12, 1463-1475. https://doi.org/10.3390/ma12091463.

Round  2

Reviewer 2 Report

 The manuscript can be accepted in present form